# Twenty-Eight Fungal Secondary Metabolites Detected in Pig Feed Samples: Their Occurrence, Relevance and Cytotoxic Effects In Vitro

**DOI:** 10.3390/toxins11090537

**Published:** 2019-09-14

**Authors:** Barbara Novak, Valentina Rainer, Michael Sulyok, Dietmar Haltrich, Gerd Schatzmayr, Elisabeth Mayer

**Affiliations:** 1BIOMIN Research Center, Technopark 1, 3430 Tulln, Austria; valentina.rainer@biomin.net (V.R.); gerd.schatzmayr@biomin.net (G.S.); e.mayer@biomin.net (E.M.); 2Institute of Bioanalytics and Agro-Metabolomics, University of Natural Resources and Life Sciences, Konrad-Lorenz-Straße 20, 3430 Tulln, Austria; michael.sulyok@boku.ac.at; 3Food Biotechnology Laboratory, Department of Food Science and Technology, University of Natural Resources and Life Sciences, Muthgasse 11, 1190 Vienna, Austria; dietmar.haltrich@boku.ac.at

**Keywords:** *Fusarium*, *Aspergillus*, *Penicillium*, *Alternaria*, fungi, emerging mycotoxin, in vitro, IPEC-J2, occurrence data

## Abstract

Feed samples are frequently contaminated by a wide range of chemically diverse natural products, which can be determined using highly sensitive analytical techniques. Next to already well-investigated mycotoxins, unknown or unregulated fungal secondary metabolites have also been found, some of which at significant concentrations. In our study, 1141 pig feed samples were analyzed for more than 800 secondary fungal metabolites using the same LC-MS/MS method and ranked according to their prevalence. Effects on the viability of the 28 most relevant were tested on an intestinal porcine epithelial cell line (IPEC-J2). The most frequently occurring compounds were determined as being cyclo-(L-Pro-L-Tyr), moniliformin, and enniatin B, followed by enniatin B1, aurofusarin, culmorin, and enniatin A1. The main mycotoxins, deoxynivalenol and zearalenone, were found only at ranks 8 and 10. Regarding cytotoxicity, apicidin, gliotoxin, bikaverin, and beauvericin led to lower IC_50_ values, between 0.52 and 2.43 µM, compared to deoxynivalenol (IC_50_ = 2.55 µM). Significant cytotoxic effects were also seen for the group of enniatins, which occurred in up to 82.2% of the feed samples. Our study gives an overall insight into the amount of fungal secondary metabolites found in pig feed samples compared to their cytotoxic effects in vitro.

## 1. Introduction

Mycotoxins are described as secondary fungal metabolites produced by filamentous fungi in the field, and during storage and transportation under appropriate conditions. The main fungal species *Fusarium*, *Alternaria*, *Aspergillus*, and *Penicillium* possess a remarkable potential to produce a wide range of metabolites. Researchers assume that around 38% of the known 22,630 microbial products are of fungal origin; however, the exact number is the still subject of research [1].

The European Food Safety Authority (EFSA) set maximum allowed concentration levels or guidance values for some of these metabolites called “regulated mycotoxins” [2]. These include aflatoxins, deoxynivalenol, HT-2/T-2 toxins, zearalenone, fumonisins, and ochratoxin A, which have been proven to cause detrimental effects in humans and animals. Some other fungal metabolites are so-called “emerging mycotoxins” and are claimed to be metabolites for which no guidance values exist yet. Scientific opinions and risk assessments have only been published for aflatoxin precursors (such as sterigmatocystin), ergot alkaloids, enniatins, beauvericin, and moniliformin so far [3,4,5,6]. However, for many other fungal metabolites, only a few, contradictory studies are available [7,8,9]. One reason for the underestimation of their potential threat might be the limited knowledge about their prevalence. Because of the continuous development of LC-MS/MS (liquid chromatography tandem mass spectroscopy) methods, a more sensitive detection of a wide range of secondary fungal metabolites in feed and food matrixes is now available [10]. Furthermore, studies indicate that climatic change might affect mycotoxin production patterns, leading to an increase in the emerging mycotoxin level [11,12]. In 2013, Streit et al. [13] published data on the *Fusarium* metabolites beauvericin, enniatins, aurofusarin, and moniliformin, which occur in 98%, 96%, 84%, and 76% of 83 analyzed feed samples, respectively. In the same study, the well-investigated mycotoxins deoxynivalenol (89%), zearalenone (87%), and fumonisins (22%) were additionally found in the analyzed samples. Besides, metabolites produced by *Alternaria* spp. were detected in many of the analyzed feed samples (e.g., alternariol, 80%; alternariol monomethyl ether, 82%; tenuazonic acid, 65%; and tentoxin, 80%). Although further publications revealed higher occurrences and concentrations of uncommon mycotoxins, they were limited to specific geographic regions and feed commodities [14,15,16,17,18]. 

The aim of this study was, first, to provide a complete picture of the effects of 28 chemically diverse fungal metabolites on the cell viability of an intestinal porcine epithelial cell line (IPEC-J2), and second, to compare the tested concentrations with the actual values found in pig feed samples worldwide. The cell line IPEC-J2 was chosen as an optimal test model since the intestine is the first target after ingesting mycotoxin-contaminated feed and most of the metabolites are absorbed in the jejunum [19]. The selection of fungal metabolites to be tested was mainly based on their relevance and occurrence in pig feed samples obtained from the BIOMIN mycotoxin survey program, conducted in the years 2014 to 2019. Here, the most relevant metabolites were *Fusarium*-derived compounds, such as enniatins, beauvericin, moniliformin, culmorin, fusaric acid, etc., but also many compounds produced by the genera *Alternaria*, *Penicillium*, and *Aspergillus* were evaluated. Cytotoxicity studies for selected metabolites are available for different cell types [9,20,21,22,23], but here we compared, for the first time, the in vitro effects of 28 metabolites taken from occurrence data of 1141 pig feed samples analyzed with the same analytical method [18].

## 2. Results

### 2.1. Occurrence Data

The challenge of collecting and analyzing manifold feed samples from all over the world is not only a logistical one, but is mainly found in trying to measure the samples with the same analytical method, limits of detection (LOD), and performance parameters at the same instrument [11] to achieve comparable results. Therefore, BIOMIN started a unique global mycotoxin survey including samples from 44 countries, obtained between February 2014 and February 2019, where 4978 feed samples were analyzed for more than 800 different metabolites using a multi-analyte LC-MS/MS-based method [24]. For our study, we selected 1141 out of the 4978 feed samples that were only intended for swine nutrition.

#### Secondary Fungal Metabolites in Finished Pig Feed Samples

Analyzed pig feed samples were taken as described in Section 5.3.2. and evaluated according to the method of Malachová et al. [24]. The list of analytes in this method has increased over recent years and covers now more than 800 metabolites. The most frequently occurring secondary fungal metabolites of 34 emerging and masked mycotoxins, as well as regulated mycotoxins in samples from 44 different countries, are summarized in Table 1. Twenty-eight metabolites, which are not regulated yet or are defined as an “emerging mycotoxin” [25], were selected to be tested for their effects on the viability of an intestinal porcine epithelial cell line (IPEC-J2). Gliotoxin and patulin were added to these tests because of their previously described harmful effects, making novel data beneficial for the scientific community [26,27]. The uncommon metabolites 15-hydroxy-culmorin, infectopyron, and asperglaucide could not be tested because authentic analytical standards were not commercially available. The prevalence (%), mean, median, and maximum concentrations (in µg/kg) for each compound are summarized in Table 1. In the following section, the focus lies only on the median and maximum concentrations for better legibility. 

A ranking according to the prevalence of the main 34 secondary fungal metabolites as shown in Table 1 indicates that the cyclic dipeptide cyclo-(L-Pro-L-Tyr) was most frequently found with an occurrence of 87.6% and a maximum concentration of 34,910 µg/kg; followed by moniliformin (82.6%) and enniatin B (82.2%). Even though the median concentrations of the latter two were only 17 and 30 µg/kg, individual samples reached levels of up to 2053 µg/kg and 1514 µg/kg, respectively. In 82.1% of all pig feed samples, enniatin B1 had similar concentrations as enniatin B (max. 1846 µg/kg; median 34 µg/kg). Two other common enniatins, enniatin A1 and A, were found in 77.4% and 49.5% of all samples, albeit at lower concentrations compared to the B analogues (B and B1). Additionally, samples were vastly contaminated with the *Fusarium* metabolites aurofusarin and culmorin, which were identified in approximately 909 feed samples (79%), with median concentrations of 210 µg/kg and 118 µg/kg, which was comparable to the median concentration of deoxynivalenol (DON) with 193 µg/kg measured in 879 feed samples (77%). This contrasts with the maximum measured concentrations found for aurofusarin (85,360 µg/kg) and culmorin (157,114 µg/kg), while only 34,862 µg/kg for DON was detected. The second regulated mycotoxin on our list was zearalenone (ZEN), which was found at rank 10 with a prevalence of 73.3% and a median concentration of 18 µg/kg. Another uncommon metabolite, which was determined in 75.4% of all samples, was tryptophol, which was thus positioned before ZEN in the ranking according to prevalence. About 69% of the samples were positive for the hexadepsipeptide beauvericin (BEA) with a median concentration of only 6 µg/kg. Individual samples were contaminated with a maximum concentration of 413 µg/kg BEA.

Mainly unexpected compounds, such as emodin, brevianamid F, equisetin (EQUI), and cyclo-(L-Pro-L-Val), were found at the prevalence ranking positions of 13, 15, 16 and 34, respectively. These were nevertheless detected in about 70, 69, 64 and 62% of all feed samples, respectively, relating to 799, 787, 730 and 707 feed samples, respectively, out of a total of 1141. Compounds found to a lesser extent were *Alternaria* metabolites, such as tenuazonic acid (55.0%), alternariol (50.7%), alternariol monomethyl ether (40.3%), and tentoxin (37.3%). Another group of detected metabolites was produced by *Aspergillus* and/or *Penicillium* spp., but also by some *Fusarium* species, and these included 3-nitropropionic acid (3-NP), apicidin (API), kojic acid, bikaverin, fusaric acid, and mycophenolic acid. While 3-NP and API were found in 56.5% and 52.2% of samples, respectively, the others were detected in ≤33.7% feed samples. 

Other regulated mycotoxins, such as fumonisin B1, ochratoxin A, and aflatoxin B1, did not occur frequently or in high concentrations in pig feed samples. These metabolites were determined in only 43% (median 82.7 µg/kg), 5% (median 2.6 µg/kg), and 3% (median 2.0 µg/kg) of all samples, respectively (data not shown). The mycotoxins gliotoxin and patulin, relevant due to their toxicity, scarcely occurred (0.2%) or could not be detected (<LOD) in these pig feed samples.

As seen in Figure 1, 77.7% of the obtained samples had their origin in Europe, mainly from Germany (22.4%), Denmark (15.3%), and Austria (14.1%). Few samples came from other parts of the world, such as Central and South America (9%), Russia and Asian countries (5.6%), North America (3.5%), and Africa (2.4%). The analyzed feed samples obtained from Australia (0.6%) were marginal. Thirteen samples (1.1%) could not be assigned to a specific country. It can thus be stated that the picture of the occurrence primarily reflected the situation of Central Europe, since no major changes were seen in the ranking when samples from the other countries were excluded from the analysis. 

### 2.2. Cell Viability after 48 h Toxin Treatment

The cell viability of intestinal porcine epithelial cells (IPEC-J2) was assessed after an incubation of 48 h with the respective metabolite in concentrations of up to 150 µM. Absolute and relative IC_50_ values were calculated. Deoxynivalenol (DON) (Figure 2A) was included in the test system as an internal standard to have comparable values representing a well-investigated mycotoxin. DON had already reduced the cell viability to 78.8% at 1.25 µM, resulting in absolute and relative IC_50_ values of 2.55 µM and 1.88 µM, respectively. The 28 tested fungal metabolites are listed according to their calculated absolute IC_50_ value, starting from the strongest (for 11 metabolites, relative and absolute IC_50_ value could be calculated) over moderate (for 5 metabolites, only a relative IC_50_ value could be calculated) to no cytotoxicity (for 12 metabolites, IC_50_ calculation was not possible). Apicidin was the metabolite that showed the strongest cytotoxic effect with an absolute IC_50_ value of 0.52 µM (relative 0.49 µM), followed by gliotoxin, bikaverin, beauvericin, and patulin. For all tested metabolites except enniatin B, the calculated relative IC_50_ value was lower than the absolute IC_50_ value (Figure 2).

The next six fungal metabolites that showed a strong cytotoxic effect are presented in Figure 3. The four enniatins B, A, B1, and A1 showed similar absolute IC_50_ values of 3.25 µM, 3.40 µM, 3.67 µM, and 4.15 µM, respectively. Aurofusarin was less toxic, resulting in an absolute IC_50_ value of 11.86 µM. Although emodin was tested at higher concentrations, viability did not further decrease at concentrations >50 µM. Hence, IC_50_ values of 18.71 µM (absolute) and 13.09 µM (relative) were calculated.

Assessment of the following five fungal metabolites resulted in moderate cytotoxicity on IPEC-J2, as relative, but no absolute, IC_50_ values could be calculated (Figure 4). Examination of the effects of equisetin was only possible up to a concentration of 20 µM, leading to a decreased viability of 64.0% (±7.6%). The other fungal metabolites seen in Figure 4 (B–E) were tested in a concentration range of 0.625 to 150 µM. The *Alternaria* metabolites tenuazonic acid (B) and alternariol (C) showed similar effects of reducing cell viability starting at 20 µM (76.2% and 76.3%), resulting in relative IC_50_ values of 20.88 µM and 20.26 µM. However, despite increased concentrations being used, viability remained around 50%. Rubrofusarin (D) reduced viability started at 50 µM (64.0 ± 8.7%) with a calculated relative IC_50_ value of 21.54 µM. Interestingly, mycophenolic acid (E) led to an abrupt decreased viability (26.1% ± 12.1%) at very low concentrations (0.156 to 1.25 µM), but no further loss in viability was determined at concentrations from 2.5 to 150 µM.

For the following twelve fungal secondary metabolites, shown in Figure 5 and Figure 6, no reduced cell viability was seen, and hence, no IC_50_ calculation was possible.

An overview of the obtained IC_50_ values (in µM and µg/kg) for the 16 secondary fungal metabolites with strong and moderate effects on viability is presented in Table 2, together with their median and maximum concentrations in µg/kg found in 1141 pig feed samples.

## 3. Discussion

Twenty-eight fungal metabolites plus DON were assessed for their effects on the viability of intestinal porcine epithelial cells (IPEC-J2) in comparison to their prevalence in 1149 analyzed pig feed samples. The strength of our study is that analytical measurements were performed by using one single LC-MS/MS multi-mycotoxin method, as well as the same cell line and viability assay, for all samples to achieve comparable data [24]. A general problem with global mycotoxin occurrence data is usually the difference in methodologies, which make a comparison of concentrations challenging due to different limits of detection (LODs), sample extractions, and performance parameters [11]. Research on toxicology and occurrence of emerging mycotoxins is still scarce, although it has been steadily rising during the past few decades. This can partly be explained by the fact that the sensitivity and potential of LC-MS-based methods have been improved by analyzing hundreds of metabolites simultaneously, as well as by lower detection limits [24]. Furthermore, tremendous climatic abnormalities in some parts of the world favor an increasing formation of uncommon fungal metabolites [12]. Even though not all the detected metabolites might be relevant regarding food and feed safety, the abundance and co-occurrence of those compounds in different feed matrixes might pose a certain risk to susceptible animals. IPEC-J2 provides an optimal in vitro model, as swine is considered as the most sensitive species regarding mycotoxicosis [28]. This non-transformed cell line is isolated out of jejunal epithelial cells, in which the absorption of nutrients and other compounds mainly takes place. Furthermore, it possesses strong morphological and functional similarity to intestinal epithelial cells in vivo [29,30] compared to cancer cell lines such as Caco-2 and HT-29 cells. For in vitro experiments, we decided to test concentrations of up to 150 µM or, in order not to exceed a solvent concentration of 1%, the highest possible test concentration. A solvent concentration of 1% did not negatively influence viability in our test system (data not shown). For our study, we have chosen to discuss the absolute IC_50_ value, as this value is representative for the concentration, where the half-maximal inhibitory concentration was calculated. As described by Sebaugh [31], when the response of more than two assay concentrations is above 50%, the calculation of the relative IC_50_ value would be ambiguous. Furthermore, the calculation of the relative IC_50_ value uses the top and bottom plateau, even when values do not reach 50% viability. Therefore, those IC_50_ values would result in false positive results. Mycophenolic acid (MPA) is a representative example, as viability was never lower than 66%; however, a relative IC_50_ value of 0.543 was calculated, making this toxin one of the most toxic in our ranking. The calculation for the absolute IC_50_ value was not possible for this toxin, and therefore, the calculation of the absolute IC_50_ values was used, first, because it was more accurate, second, only with this value were we able to rank these fungal metabolites according to their toxicity, and third, for reflecting a realistic scenario. For the sake of completeness, both IC_50_ values are shown if calculation was possible. For comparison with the occurrence data, maximum and median concentrations are used, as the median concentration seems to be a more accurate measure of central tendency because of a generally skewed data distribution.

Deoxynivalenol (DON) was included in our study as a comparable internal reference toxin since it is well investigated and manifold deleterious effects of DON are known, as summarized in the review by Pestka [32]. For DON, an IC_50_ concentration of 2.55 µM (= 756 µg/kg) was calculated, whereas a median concentration of 193 µg/kg was found in swine feed. Although the detected concentration was lower than the determined IC_50_ value, it is known that in particular chronic low doses of DON lead to immune dysregulation, growth retardation, and impaired reproduction [32]. This is of increased importance because the bioavailability for DON after oral administration is 98.6% ± 23.6% in pigs [33]. This might also be true for other metabolites, but neither feeding trial nor bioavailability studies have been conducted for most of them. Therefore, the choice of in vitro concentrations is challenging. Stability and retention time in the gastrointestinal tract (GIT) have not been sufficiently researched. Thus, an accumulation of metabolites in the GIT by ingesting chronic low concentrations is very likely and we tried to test a broad concentration range up to 150 µM, if possible.

Apicidin (API), only discovered in the year 1996 and isolated from *Fusarium* spp., showed the strongest cytotoxicity in our test system (IC_50_ of 0.52 µM = 324 µg/kg). This is in accordance with a study on other cell lines, reporting an IC_50_ concentration of 0.16 to 3.8 µM on cell proliferation [34]. Furthermore, this compound possesses antiprotozoal activity [35] and causes toxic effects in rats leading to death at levels of 0.05 and 0.1% [36]. Although the measured median concentration in feed samples was rather low (8 µg/kg), API was found in more than half of the samples and individual samples reached concentrations of up to 1568 µg/kg. Streit et al. [13] detected a maximum concentration of 160 µg/kg in 66% positively tested samples, but additional occurrence studies are missing. The second-ranked cytotoxic compound was gliotoxin (GLIO) with an IC_50_ value of 0.64 µM (= 209 µg/kg). This *Aspergillus fumigatus* metabolite hardly occurs in swine feed, but was detected in corn silage used as cattle feedstuff [37]. GLIO has been reported to cause immunosuppressive, genotoxic, apoptotic, and cytotoxic effects determined in a rat intestinal cell line (IEC-6), in hamster ovary cells (CHO), and in mouse macrophages (RAW264.7) [38,39], and might pose a risk for other animal species that are frequently exposed to this mycotoxin. Literature about the effects and occurrence of our third-ranked toxic compound bikaverin (BIK) is scarce. It is reported as a red pigment produced by different *Fusarium* spp. with antibiotic and antibacterial properties [40,41]. A study from 1975 reported its cytotoxicity against three different cancer cell lines, leading to ED_50_ values of 0.5–4.2 µg/mL (1.31–10.99 µM) [42], which is comparable with our IC_50_ value of 1.86 µM. Furthermore, we found BIK in almost 30% of the feed samples with a maximum concentration of 1564 µg/kg, whereas another study reported contaminations of up to 51,130 µg/kg [13]. Our in vitro data give the first evidence of its cytotoxic effects on epithelial cells, and considering an IC_50_ value of only 1.86 µM (= 711 µg/kg), the high contamination level might be alarming. We would like to point out that for this toxin, viability did not further decrease below 50% after applying higher concentrations, which has also been observed for other metabolites such as enniatin B, emodin, and tenuazonic acid. More research has been carried out for the cyclic hexadepsipeptides beauvericin (BEA) and the group of enniatins (ENNs; A, A1, B, B1). Their cytotoxicity was demonstrated in a variety of cell culture models [7,9,43,44,45,46] and is attributed to their ionophoric properties. A recently published study from Fraeyman et al. [46] revealed that proliferating IPEC-J2 are more sensitive to BEA, but not to the ENNs. Differentiated cells seem to be more robust against their detrimental effects, which was already shown by Springler and Broekaert et al. [47,48]. According to Fraeyman’s study, the overall cytotoxicity after 24 h of incubation was ranked as BEA > ENN A >> ENN A1 > ENN B1 >>> ENN B. This contrasts with our results that led to the following ranking: BEA > ENN B > ENN A > ENN B1 > ENN A1 in proliferating IPEC-J2. This discrepancy could be due to different incubation time points (24 h vs. 48 h) and different measured endpoints (flow cytometry vs. Sulforhodamine B assay). Additionally, more data about their occurrence are available. We mostly found ENN B and B1 (≈82% positive); however, 77.4%, 68.7%, and 49.5% of the feed samples were also contaminated with ENN A, BEA, and ENN A1, respectively. The maximum detected concentrations varied between 1514, 1846, 307, 413 and 549 µg/kg, respectively. Considering that those compounds usually co-occur, the total amount can exceed a level of 4500 µg/kg or even 5543 µg/kg, as described in Kovalsky et al. [18]. A high prevalence of BEA and ENNs in cereal grains was already described in other peer-reviewed studies [14,15,18,49]. Additionally, a high oral bioavailability, particularly seen for ENN B1, has been reported [50,51], which turn them into a potential risk factor for exposed animals. 

For patulin (PAT), a low IC_50_ value of 3.21 µM (= 495 µg/kg) was determined as well, but since not a single sample was contaminated with PAT, it does not seem to be relevant regarding porcine health. PAT is better known as a feed contaminant in fruits, especially apples and vegetables, and its toxicity was demonstrated in different animal species [52,53]. Other conclusions have to be drawn regarding the *Fusarium* metabolite aurofusarin (AURO), which led to an IC_50_ value of 11.86 µM (= 6766 µg/kg) and was detected in a median and maximum concentration of 210 µg/kg and 85,360 µg/kg, respectively. An interference of AURO with antioxidants and fatty acids in the eggs and embryos of quails has already been reported [54,55], as well as its negative effect on the growth performance in red tilapia [56] and cytotoxicity in mammalian cells [23,57,58]. Our results are comparable with a study from Jarolim et al. [23], reporting a significant decrease in viability of HT29 and HCEC-1CT cells starting at 5 µM AURO after 48 h. Even though the calculated IC_50_ value of 6766 µg/kg seems to be high in our approach, when considering the maximum found concentration of 85,360 µg/kg, a critical risk assessment is urgently required. The last metabolite where an IC_50_ calculation was possible was emodin (EMO), which is claimed to be a therapeutic agent of various diseases used in traditional Chinese medicine for centuries. As EMO is produced by a range of different plant families, and found ubiquitously in herbs, trees, and shrubs [59], a potential risk to animal health seems very unlikely. 

Five out of the tested metabolites, mycophenolic acid (MPA), equisetin (EQUI), alternariol (ALT), tenuazonic acid (TeA), and rubrofusarin (RUB) led to a slight decrease in viability, and relative IC_50_ values could be determined. Despite increasing concentrations, a reduction of more than 50% was not found; therefore, absolute IC_50_ values could not be calculated. We have chosen the SRB assay to measure the protein content because this cell target was the most sensitive one in preliminary studies. However, it seems that for other compounds, a different cell target might be more suitable. As described in an study from 2017 [60], different endpoint analyses can lead to a different outcomes. Especially for MPA, a study about its cytotoxicity has been published where the mitochondrial activity of Caco-2 cells was only decreased by 45% at 780 µM MPA after 48 h of incubation, but no effect was seen in THP-1 monocytes [61]. These results vary from ours since we determined a constant decline of 20.0 to 33.6% viability already starting from 1.25 to 150 µM MPA by measuring the cellular protein content. Interestingly, similar results about ALT were published, in which the cell viability of HepG2 cells did not decrease in a concentration-dependent manner up until a concentration of 100 µM [62,63].

None of the other compounds, culmorin (CUL), moniliformin (MON), roquefortine C, tentoxin, alternariol monomethyl ether, kojic acid, cyclo-(L-Pro-L-Tyr), cyclo-(L-Pro-L-Val), tryptophol, 3-nitropropionic acid, brevianamid F, and fusaric acid (FA) showed negative effects in the tested concentration range on IPEC-J2. However, we would like to refer to four of them: although the metabolite culmorin (CUL) elucidated no cytotoxic effect in our tests, it was found in 79.7% of all samples, with remarkable concentrations of 157,114 µg/kg. Interestingly, its natural occurrence is always correlated to the occurrence of DON. A recent study showed that CUL is able to inhibit the glucuronidation of DON in human liver microsomes, and thus, its detoxification process [64,65]. These findings make the compound highly relevant regarding synergistic effects, not only for the detoxification of DON, but also for other toxins. MON was already described as a hazardous contaminant for poultry, and therefore, a risk assessment has been recently published by EFSA [6,66]. Finally, although being non-cytotoxic in our experimental approach, cyclo-(L-Pro-L-Tyr) was found to be the most frequently detected compound in our survey program with 87.6% positive samples. Only a few publications described its antibacterial activity against two gram-negative bacteria, as well as its cytotoxic and genotoxic effects in lymphocytes and various cell lines to date [67,68,69]. However, due to its high occurrence, along with its high concentration, further studies are needed. Furthermore, we would like to stress that other metabolites, such as aflatoxin B1 and fumonisin B1, are also known for their non-cytotoxic effects in vitro and their detrimental effect in animals. Thus, a lack of cytotoxicity in vitro does not necessarily indicate a complete harmlessness.

## 4. Conclusions

Taken together, our study focused on the occurrence data and concentrations of 28 secondary fungal metabolites without regulatory guidelines and their effect on the viability of an intestinal porcine cell line. In the majority of the analyzed pig feed samples, low median concentrations (15 of 28 metabolites gave median concentrations of <20 µg/kg) were determined; however, some individual samples were contaminated with high concentration levels, which might be relevant for animal health. The maximum occurrence values exceeded the absolute IC_50_ concentrations for apicidin, bikaverin, and aurofusarin. Moreover, even if acute exposure to most of the metabolites is low, concerns regarding chronic exposure at lower levels are rising. An important factor that needs to be considered for further investigations comprise the absorption, distribution, metabolism, and excretion (ADME) of the substances and their ability to enter the target cell. Therefore, cytotoxicity studies provide a first overall picture of the relevance of the detected compounds and serve as a suitable alternative and prerequisite to animal testing. For a qualified risk assessment, however, reliable combination studies to investigate synergistic, additive, and antagonistic effects are needed due to the frequent co-occurrence of toxic compounds, especially with the regulated main mycotoxins. Finally, data from feeding trials in productive livestock with chronic exposure of those compounds have to be the logical target for the testing process within the next few years.

## 5. Materials and Methods 

### 5.1. Cell Culture

The porcine jejunal intestinal epithelial cell line, IPEC-J2 (ACC 701; Leibnitz Institute DSMZ, German Collection of Microorganisms and Cell Cultures, Braunschweig, Germany), originated from a neonatal, unsuckled piglet. These non-transformed cells were continuously maintained in complete cultivation medium consisting of Dulbecco’s modified eagle medium (DMEM/Ham’s F12 (1:1), Biochrom AG, Berlin, Germany), supplemented with 5% fetal bovine serum, 1% insulin-transferrin-selenium, 5 ng/mL epidermal growth factor, 2.5 mM Glutamax (all GibcoTM, Life Technologies, Carlsbad, CA, USA), and 16 mM 4-(2-hydroxyethyl)-1-piperazineethanesulfonic acid (Sigma-Aldrich, St. Louis, MO, USA), and grown at 39 °C in a humidified atmosphere of 5% CO_2_. IPEC-J2 between passages 1–15 were routinely seeded at 1 × 10^6^ cells/mL in 150 cm^2^ tissue culture flasks (Starlab, Hamburg, Germany) with 28 mL complete cultivation medium and subcultured upon confluence every 3–4 days. For assays, confluent cells were detached using Trypsin (0.25%)-ethylenediaminetetraacetic acid (EDTA, 0.5 mM, Sigma-Aldrich, St. Louis, MO, USA). Cell culture was regularly tested and found to be free of mycoplasma contamination via PCR (Venor® GeM Mycoplasma Detection Kit; Minvera Biolabs GmbH, Berlin, Germany). 

### 5.2. Material

All tested chemicals are listed in Table 3 and were dissolved either in dimethylsulfoxid (DMSO, Sigma-Aldrich, St. Louis, MO, USA), ethanol (EtOH absolut, VWR International, Radnor, PA, USA), acetonitrile (ACN, Sigma-Aldrich, St. Louis, MO, USA), or distilled water.

### 5.3. Method

#### 5.3.1. Cell Viability Assay

For the measurement of cell viability, 3 × 10^4^ cells/well were seeded into a flat-bottom, cell-culture-treated 96-well microplate (Eppendorf) in 200 µL cultivation media and grown for 24 hours at 39 °C and 5% CO_2_. After reaching confluence, IPEC-J2 were treated with a broad range of concentrations of all chemicals (listed in Table 3).

A sulforhodamine B (SRB) assay (Xenometrix, Allschwil, Switzerland) was used to determine cellular protein content and was performed according to the manual. Briefly, the supernatant was discarded and the cell layer was washed with 300 µL SRB I solution per well. Then, 100 µL SRB II fixing solution was added to each well and the plate was incubated for 1 h at 4 °C. After the incubation time, cells were washed three times with 200 µL distilled water. Cells were stained with 50 µL SRB III labelling solution/well for 15 min, and afterwards, cells were washed again two times with 400 µL SRB IV rinsing solution. Then, bound SRB was solubilized with 200 µL SRB V and after 15–60 min, absorbance was measured at 540 nm and a reference wavelength of 690 nm using a microplate reader (Synergy HT, Biotek, Bad Friedrichshall, Germany).

#### 5.3.2. LC-MS/MS Multi-Analyte Method

A total of 1141 samples from 75 countries were provided by the BIOMIN Mycotoxin Survey for measuring up to 800 analytes, including fungal and bacterial metabolites, pesticides, and veterinary drug residues using a LC-MS/MS multi-mycotoxin analysis method [24]. Samples were collected after instruction or by trained staff only from February 2014 until February 2019. For the present study, data from finished pig feed samples including 28 fungal metabolites were chosen for detailed analysis (see Table 1). The threshold *(t)* was set to be >1.0 µg/kg or the limit of detection (LOD), whichever was higher. 

A minimum of 500 g of sample was submitted to the laboratory of the Institute of Bioanalytics and Agro-Metabolomics at the University of Natural Resources and Life Sciences Vienna (BOKU) in Tulln. After reception, samples were immediately milled, homogenized, and finally analyzed. Samples were extracted with a mixture of acetonitrile (ACN), water, and acetic acid (79:20:1, per volume) on a rotary shaker for 90 min. After centrifugation, the supernatant was transferred to a glass vial and diluted with a mixture of ACN, water, and acetic acid (20:79:1, per volume), and was injected into the LC-MS/MS system (electrospray ionisation and mass spectrometric detection employing a quadrupole mass filter). Quantification was done according to an external calibration using a multi-analyte stock solution.

The method was performed according to the guidelines from the Directorate General for Health and Consumer affairs of the European Commission, published in document No. 12495/2011 [70]. 

#### 5.3.3. Statistics and Evaluation

The half-maximal inhibitory concentrations (IC_50_) were calculated using GraphPad Prism (GraphPad Prism Version 7.03, San Diego, CA, USA). For calculation of the relative IC_50_ value, data was log-transformed and fitted to a four-parameter logistic equation:(1)Y=Bottom+(Top−Bottom)/(1+10((LogIC50−X) x Hillslope

The molar concentration of a substance that reduced viability to 50% between the top and the bottom plateau was calculated.

For the calculation of the absolute IC_50_ value, data was log-transformed and following equation was used: (2)$$Y=Bottom+\left(Top-Bottom\right)/(1+10\left(\left(LogIC50-X\right)\;x\;HillSlope+\log\left(\frac{Top-Bottom}{Fifty-Bottom}-1\right)\right)$$

Fifty = 50; Top = 100

The molar concentration of a substance that reduced viability to 50% of the maximum viability was calculated.

## Figures and Tables

**Figure 1 toxins-11-00537-f001:**
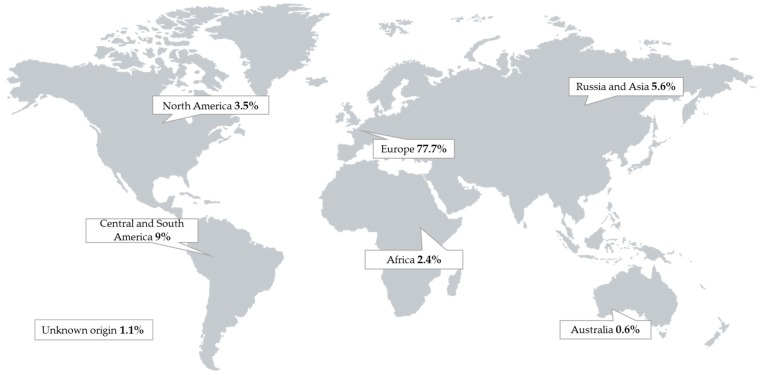
Origin of 1141 pig feed samples obtained from different parts of the world.

**Figure 2 toxins-11-00537-f002:**
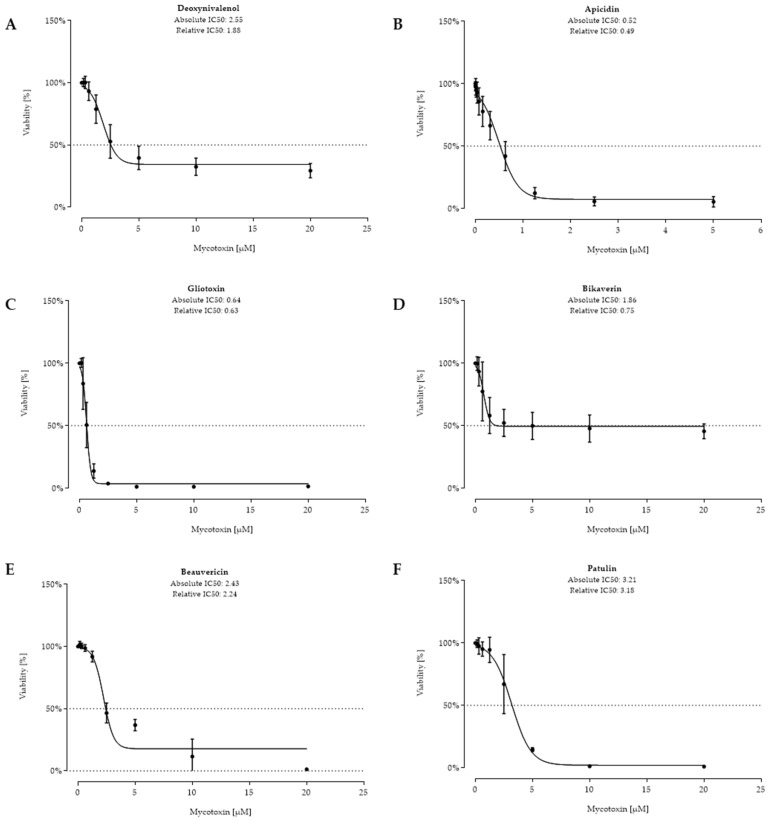
Viability (%) after 48 h of incubation with deoxynivalenol (**A**), apicidin (**B**), gliotoxin (**C**), bikaverin (**D**), beauvericin (**E**), and patulin (**F**) (tested at (0.156–20 µM)); except for apicidin (0.0049–5 µM)) of confluent intestinal porcine epithelial cells (IPEC-J2). Data represent mean ± standard deviation.

**Figure 3 toxins-11-00537-f003:**
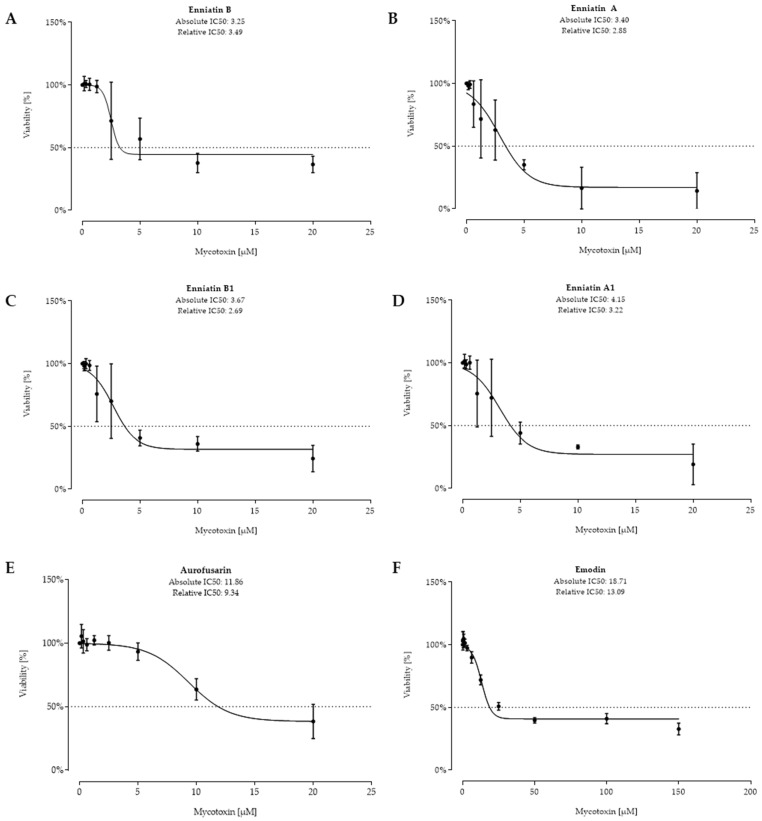
Viability (%) after 48 h of incubation of enniatin B (**A**), enniatin A (**B**), enniatin B1 (**C**), enniatin A1 (**D**), aurofusarin (**E**), and emodin (**F**) (tested at (0.156–20 µM), except for emodin (0.625–150 µM)) of confluent intestinal porcine epithelial cells (IPEC-J2). Data represent mean ± standard deviation.

**Figure 4 toxins-11-00537-f004:**
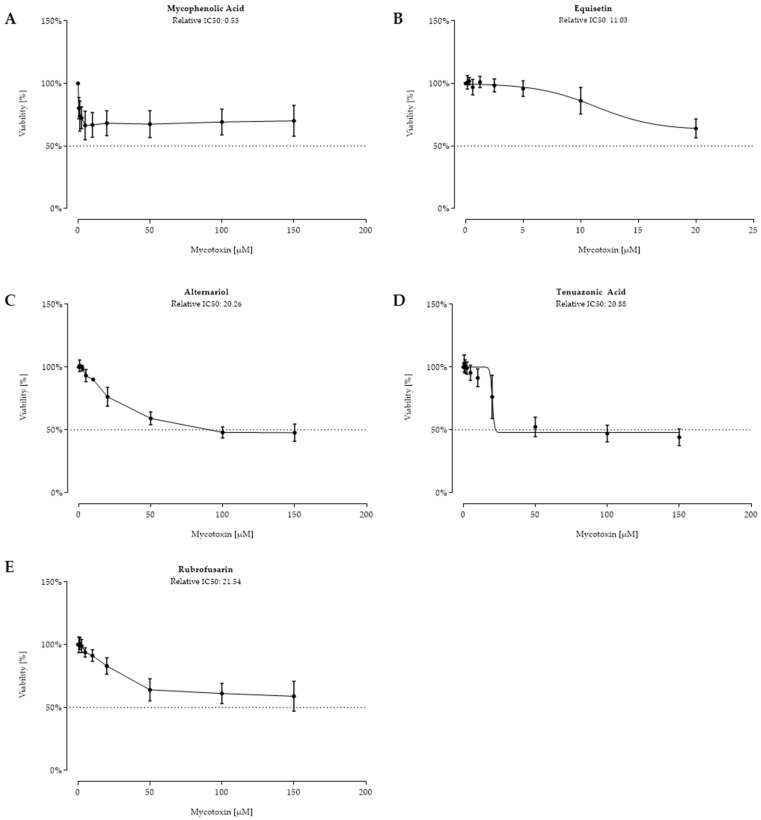
Viability (%) after 48 h of incubation with mycophenolic acid (**A**) (0.156–150 µM), equisetin (**B**) (0.156–20 µM), alternariol (**C**), tenuazonic acid (**D**), and rubrofusarin (**E**) ((0.156–150 µM) for C–E) of confluent intestinal porcine epithelial cells (IPEC-J2). Data represent mean ± standard deviation.

**Figure 5 toxins-11-00537-f005:**
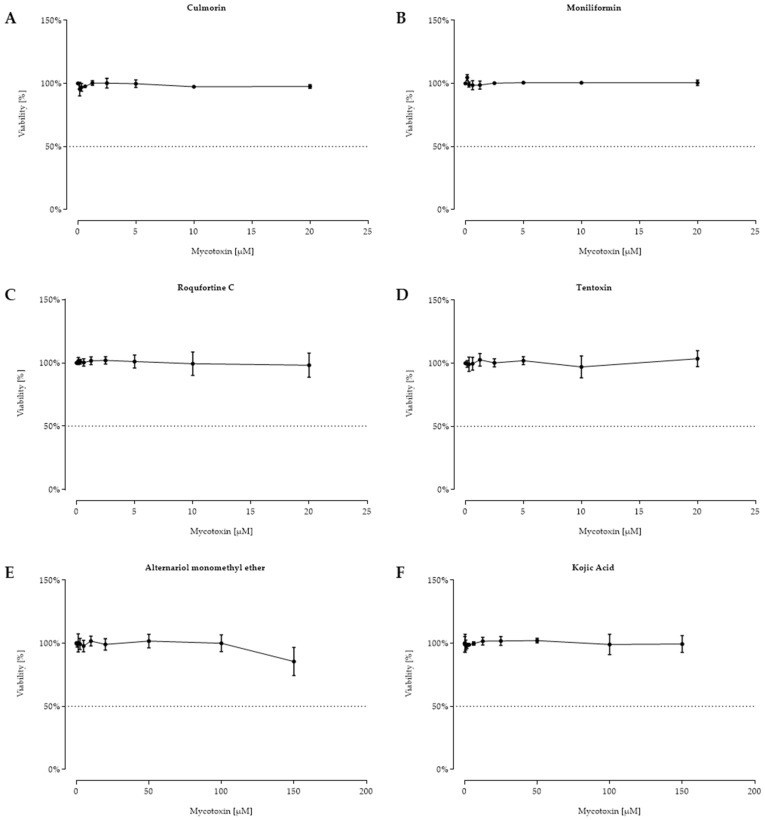
Viability (%) after 48 h of incubation with culmorin (**A**), moniliformin (**B**), roquefortine C (**C**), tentoxin (**D**) ((0.156–20 µM) for A–D), alternariol monomethyl ether (**E**), and kojic acid (**F**) ((0.625–150 µM) for E–F) of confluent intestinal porcine epithelial cells (IPEC-J2). Data represent mean ± standard deviation.

**Figure 6 toxins-11-00537-f006:**
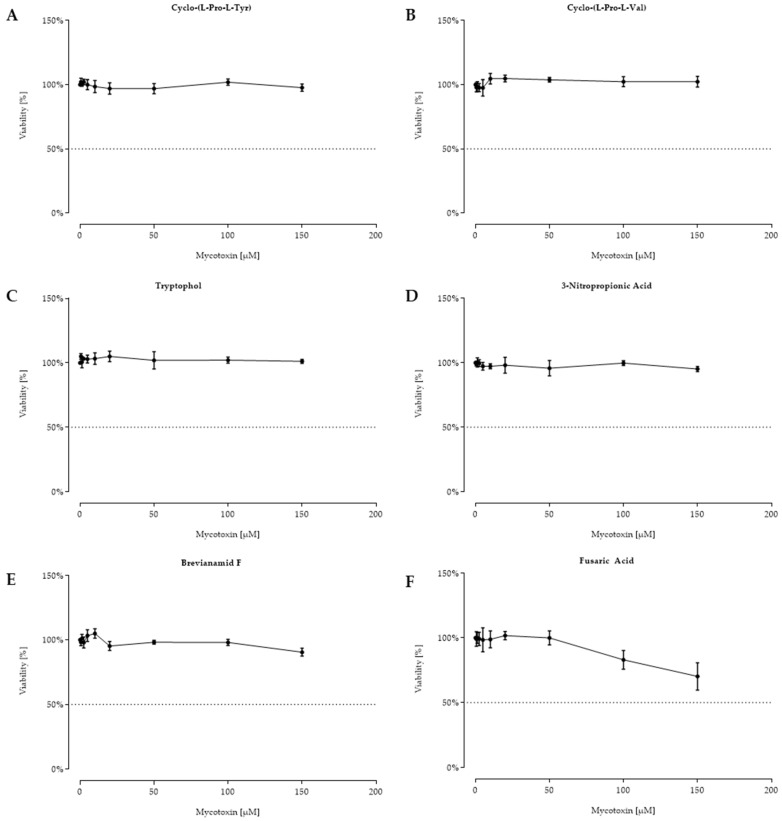
Viability (%) after 48 h of incubation with cyclo-(L-Pro-L-Tyr) (**A**), cyclo-(L-Pro-L-Val) (**B**), tryptophol (**C**), 3-nitropropionic acid (**D**), and brevianamid F (**E**), and fusaric acid (**F**) ((0.625–150 µM) for A–F) of confluent intestinal porcine epithelial cells (IPEC-J2). Data represent mean ± standard deviation.

**Table 1 toxins-11-00537-t001:** Summary of 34 secondary fungal metabolites ranked according to their prevalence with mean, median, and maximum concentration, measured in 1141 finished feed samples for swine obtained from February 2014 to February 2019 using a multi-analyte LC-MS/MS-based method.

Rank	Metabolite	Mean Concentration	Median Concentration	Maximum Concentration	Prevalence
1	Cyclo-(L-Pro-L-Tyr)	321	105	34,910	87.6%
2	Moniliformin, *t* > 2.0 ppb	66	17	2053	82.6%
3	Enniatin B	73	30	1514	82.2%
4	Enniatin B1	78	34	1846	82.1%
5	Aurofusarin	932	210	85,360	80.7%
6	Culmorin	905	118	157,114	79.7%
7	Enniatin A1	30	14	549	77.4%
8	Deoxynivalenol, *t* > 1.5 ppb	634	193	34,862	77.0%
9	Tryptophol	291	138	10,270	75.4%
10	Zearalenone*	126	18	9905	73.3%
11	15-Hydroxy-culmorin*	468	152	19,320	73.2%
12	Beauvericin, *t* > 2.0 ppb	17	6	413	68.7%
13	Emodin	17	4	591	69.3%
14	Infectopyron*	983	294	66,094	65.9%
15	Brevianamid F	44	25	1170	65.2%
16	Equisetin	50	11	6120	64.2%
17	Cyclo-(L-Pro-L-Val)	187	71	5042	62.1%
18	DON-3-glucoside*, *t* > 15.0 ppb	74	22	2741	62.8%
19	Asperglaucide*	113	31	6232	61.8%
20	Nivalenol*, *t* > 15.0 ppb	65	31	1143	56.7%
21	3-Nitro-propionic acid	16	6	509	56.5%
22	Tenuazonic acid	255	82	9910	55.0%
23	Apicidin	22	8	1568	52.2%
24	Alternariol	17	4	2508	50.7%
25	Enniatin A	7	3	307	49.5%
26	Alternariol monomethyl ether	6	3	208	40.3%
27	Tentoxin	8	3	157	37.3%
28	Kojic acid	192	78	3030	33.7%
29	Bikaverin	58	19	1564	29.8%
30	Fusaric acid	333	81	5566	13.0%
31	Mycophenolic acid	39	8	1178	13.1%
32	Rubrofusarin	199	38	1696	2.3%
33	Gliotoxin	5	5	6	0.2%
34	Patulin	<LOD	<LOD	<LOD	n.a.

Concentrations in µg/kg. * = not tested in vitro (lack of availability or known as regulated or masked mycotoxin). If not otherwise stated, a threshold (t) of >1.0 µg/kg or >LOD (limit of detection) was established.

**Table 2 toxins-11-00537-t002:** List of absolute and relative IC_50_ values in µM (left columns) and in µg/kg (middle columns), compared to median and maximum concentration in µg/kg (right columns) ranked according their cytotoxicity.

Rank	Fungal Metabolite	IC_50_ Value (µM)		IC_50_ Value (µg/kg)		Occurrence	
		Absolute	Relative	Absolute	Relative	Median (µg/kg)	Maximum (µg/kg)
1	API	0.52	0.49	324	306	8	1568
2	GLIO	0.64	0.63	209	206	5	6
3	BIK	1.86	0.75	711	287	19	1564
4	BEA	2.43	2.24	1905	1756	6	413
Control	DON	2.55	1.88	756	557	193	34,862
5	PAT	3.21	3.18	495	490	<LOD	<LOD
6	EnnB	3.25	3.49	2079	2233	30	1514
7	EnnA	3.40	2.88	2319	1964	3	307
8	EnnB1	3.67	2.69	2400	1759	34	1846
9	EnnA1	4.15	3.22	2772	2151	14	549
10	AUR	11.86	9.34	6766	5329	210	85,360
11	EMO	18.71	13.09	5056	3537	4	591
12	MPA	nc	0.53	nc	170	8	1178
13	EQUI	nc	11.03	nc	4120	11	6120
14	ALT	nc	20.26	nc	5232	4	2508
15	TeA	nc	20.88	nc	4761	82	9910
16	RUB	nc	21.54	nc	5865	38	1696

nc = not calculable.

**Table 3 toxins-11-00537-t003:** List of chemicals.

Chemical	Purity	Solvent	Highest Tested Concentration (µM)	Company
Alternariol (*Alternaria* sp.)	≈96%	DMSO	150	Sigma-Aldrich
Alternariol monoethyl ether (*Alternaria alternata)*	≥98%	DMSO	150	Sigma-Aldrich
Apicidin (*Fusarium* sp.)	≥95%	DMSO	5	Santa Cruz
Aurofusarin (*Fusarium graminearum)*	≥97%	DMSO	20	AdipoGen Life Sciences
Beauvericin (*Beauveria* sp.)	≥95%	DMSO	20	AdipoGen Life Sciences
Bikaverin (*Fusarium* sp.)	95%	DMSO	20	Santa Cruz
Brevianamid F	>98%	DMSO	150	MedChem Express
Cyclo(L-Pro-L-Tyr)	>98%	DMSO	150	BioAustralis
Cyclo(L-Pro-L-Val)	>98%	DMSO	150	BioAustralis
Culmorin	100%	ACN	20	Generous gift from Dr. Fruhmann
Deoxynivalenol (*Fusarium* sp.)	≥95%	Distilled water	20	Biopure
Emodin	≥97%	DMSO	150	Sigma-Aldrich
Enniatin A, A1, B, B1 (*Gnomonia errabunda)*	≥95%	DMSO	20	Sigma-Aldrich
Equisetin (*Fusarium equiseti)*	>99%	DMSO	20	Santa Cruz
Fusaric acid (*Gibberella fujikuroi)*	≥98%	96% EtOH	150	Sigma-Aldrich
Gliotoxin (*Gladiocladium fimbriatum)*	≥97%	DMSO	20	Santa Cruz
Kojic acid	≥99%	Distilled water	150	Sigma-Aldrich
Moniliformin (*Fusarium moniliforme)*	≥99%	Distilled water	20	BioAustralis
Mycophenolic acid (*Penicillium brevicompactum)*	≥98%	DMSO	150	Sigma-Aldrich
Patulin	98%	DMSO	20	Santa Cruz
Roquefortine C (*Penicillium* sp.)	≥98%	DMSO	20	AdipoGen Life Sciences
Tentoxin (*Alternaria tenuis*)	≥95%	70% EtOH	20	Sigma-Aldrich
Tenuazonic acid (*Alternaria alternata)*	≥98%	DMSO	150	AdipoGen Life Sciences
Tryptophol	>98%	DMSO	150	AdipoGen Life Sciences
3-Nitropropionic acid	≥97%	70% EtOH	150	Sigma-Aldrich

Adipogen Life Sciences, Switzerland; BioAustralis, Australia; Biopure, Austria; MedChem Express, Sweden; Santa Cruz, Germany; Sigma-Aldrich, USA.

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
