# Peer review of "Twenty-Eight Fungal Secondary Metabolites Detected in Pig Feed Samples: Their Occurrence, Relevance and Cytotoxic Effects In Vitro"

_toxins, 2019, doi:10.3390/toxins11090537_

Round 1

Reviewer 1 Report

The manuscript concerns to a study that evaluates, in vitro, the cytotoxic effects of main non-regulated mycotoxins found in pig feed. The work was very well conducted and present relevant data. The problem of emerging, or non-regulated mycotoxins, is very pertinent, especially when they are co-occurring with regulated mycotoxins. The manuscript is well written and needs only minor corrections.

- I question if the “1,141 pig feed samples” is essential in the title? Maybe, “pig feed samples” will be accurate enough.

- I also recommend to add “cytotoxic effects in vitro” rather than only “effects in vitro”.

- I also think that it will be better to use international units instead of ppb. This would benefit the comparison of disclosed data with other published articles. Authors only need to change from ppb to µg/L.

- I also recommend to adjust tables and text to the entire page space. Perhaps this is more dependent of journal editors than authors, but this should be improved to enhance the reading.

- Red text should be changed to black.

- The quality of images from figures should be improved to guaranty a clear readiness of text in there.

- In line 259, authors should check the sentence and see if “absolute IC50” is really what they intended to say. Did you mean “relative IC50”??

- In line 267, I recommend authors to compare the DON results “IC50 concentration of 2.55 µM (= 756 ppb)” with other published works to evaluate if those values are in agreement with other experiments.

- The same is recommended for gliotoxin in line 285, if any

-  In line 313, I recommend to compare the total concentration reported in that reference with the values detected in your samples. That is, it would be also interesting if you disclose the co-occurrence of detected metabolites in each sample. For example, are the regulated mycotoxins co-occurring with any particular non-regulated one in pig samples tested? For example, DON and enniatins are they occurring in the same samples or not? DON and any other non-regulated mycotoxin are co-occuring? This would be a valuable information for safety issues, because of toxicological addition aspects.

- In line 340, the reported MPA studies are they different from yours?

- Line 344, “kojic acid.” should be “kojic acid,”.

- Line 401, should be Table 3 and not Table 1.

- Brevianamid F, should be Brevianamide F?? This is the same of cyclo-(L-Trp-L-Pro)? Please state so.

- “Equisetin from Fusarium equiseti” should be “Equisetin from (Fusarium equiseti)“.  

- I think it would be beneficial if authors state the main origin of each metabolite. I mean, are they fungal, bacterial or plant metabolites? Are they all mainly produced by fungi? This can be easily added in table 3. In particular, tentoxin, Tryptophol and 3-Nitropropionic acid.

- Please see attached document for reference corrections and other minor rectifications.

Author Response

Dear Reviewer 1,

Thanks a lot for reading our manuscript carefully and giving us useful and valuable suggestions for improvement. We really appreciate your effort and we will refer to your comments in detail in the following:

- I question if the “1,141 pig feed samples” is essential in the title? Maybe, “pig feed samples” will be accurate enough.

I also recommend to add “cytotoxic effects in vitro” rather than only “effects in vitro”.

-  I also think that it will be better to use international units instead of ppb. This would benefit the comparison of disclosed data with other published articles. Authors only need to change from ppb to µg/L.

Thanks for the suggestions – we have changed the title without using the amount of samples and with “cytotoxic” before effects in vitro. Furthermore, we have changed all ppb concentrations to the international units µg/kg.

- I also recommend to adjust tables and text to the entire page space. Perhaps this is more dependent of journal editors than authors, but this should be improved to enhance the reading.

Yes, we have adjusted the tables.

- Red text should be changed to black.

Red text was changed.

- The quality of images from figures should be improved to guaranty a clear readiness of text in there.

We changed the quality of the figures by providing them directly out of GraphPad Prism with a higher quality of 600 dpi of resolution (instead of 300 dpi). We will not include them in the revised manuscript, but sent them directly to the editors of the journal as supplement. Thus, they can insert them in the higher quality format.

- In line 259, authors should check the sentence and see if “absolute IC50” is really what they intended to say. Did you mean “relative IC50”??

No, we really meant the absolute IC50 value, not the relative one. Details about those values are explained at the beginning of the discussion.

- In line 267, I recommend authors to compare the DON results “IC50 concentration of 2.55 µM (= 756 ppb)” with other published works to evaluate if those values are in agreement with other experiments.

Thanks for this comment. A direct comparison with other publications is rather difficult, since different cell types (cell lines, primary cells), different incubation times and different assays for measuring cell viability were used. Furthermore, not all studies refer to an IC50 values and use different methods to calculate the cytotoxicity. Since the focus of our study is not on deoxynivalenol (DON), but on the comparison of this known cytotoxic mycotoxin to other less-studied compounds, we decided not to discuss this aspect in detail. However, a broad range of studies about the detrimental effects of DON is well summarized in the reviews provided by Sobrova et al. (Deoxynivalenol and its toxicity. Interdiscip Toxicol., 2010, 3(3): 94–99.) and by Pinton and Oswald (Effect of Deoxynivalenol and Other Type B Trichothecenes on the Intestine: A Review. Toxins (Basel), 2014, 6(5): 1615–1643.

- The same is recommended for gliotoxin in line 285, if any

In our manuscript, we mentioned that gliotoxin was already shown to possess genotoxic, immunosuppressing, apoptotic and cytotoxic effects. For more information, we added the different used cell lines into the text (see line 292-293), which were a rat intestinal cell line (IEC-6), hamster ovary cells (CHO) and mouse macrophages (RAW264.7). To our knowledge, there are recently no further published data about in vitro effects of gliotoxin.

-  In line 313, I recommend to compare the total concentration reported in that reference with the values detected in your samples. That is, it would be also interesting if you disclose the co-occurrence of detected metabolites in each sample. For example, are the regulated mycotoxins co-occurring with any particular non-regulated one in pig samples tested? For example, DON and enniatins are they occurring in the same samples or not? DON and any other non-regulated mycotoxin are co-occuring? This would be a valuable information for safety issues, because of toxicological addition aspects.

We added the total maximum concentrations of enniatin A, A1, B, B1 (ENNs) and beauvericin (BEA) from the cited publication (Kovalsky et al., 2016) to our manuscript for a better comparison (see line 318). Since our study focus on the single effect, we do not go in detail regarding the co-occurrence with regulated and unregulated mycotoxins. However, we have seen from our survey program that DON primarily co-occurs with culmorin (and its hydroxy forms) and with aurofusarin in even high concentrations. Those compounds are mainly produced by the same Fusarium strains. DON also frequently co-occur with ENNs and BEA, and especially latter co-occur with each other, because they belong to the same class of compounds called cyclic depsipeptides. In general, the majority of the samples are contaminated by more than 10 metabolites, therefore, co-occurrence is most likely and determination of synergistic effects of those co-occurring mycotoxins will be our focus for further studies. Nevertheless, since our manuscript has already reached a certain length, the discussion about co-occurrence of regulated and non-regulated mycotoxins will be covered soon in an own publication which will be most likely submitted with the end of this year from one of our collaborations.

- In line 340, the reported MPA studies are they different from yours?

We added more information about the cited publication to this part, found in line 346.

- Line 344, “kojic acid.” should be “kojic acid,”.

- Line 401, should be Table 3 and not Table 1.

Thanks for reading carefully, we have changed those typos.

- Brevianamid F, should be Brevianamide F?? This is the same of cyclo-(L-Trp-L-Pro)? Please state so.

Yes, brevianamid F can also be called cyclo-(L-Trp-L-Pro). To our knowledge, the more common name is brevianamid F, therefore we would like to name it like that.

- “Equisetin from Fusarium equiseti” should be “Equisetin from (Fusarium equiseti)“.  

Thanks, we have changed that.

- I think it would be beneficial if authors state the main origin of each metabolite. I mean, are they fungal, bacterial or plant metabolites? Are they all mainly produced by fungi? This can be easily added in table 3. In particular, tentoxin, Tryptophol and 3-Nitropropionic acid.

We have added the producer from tentoxin, which was provided by Sigma. For the other two metabolites, the supplier did provided any further information about the production strains. However, fungi from Alternaria species mainly produce both compounds.

- Please see attached document for reference corrections and other minor rectifications.

Thanks again for reading the references carefully. We have added the pages in the references 3, 4, 6 and 18. Furthermore, we have changed the fungi species to italics in references 37 and 38. We reviewed reference 17, which is correctly written the way it is. For the citations 19, 43 and 47, the stated numbers are the pages, which are provided for the referred publications.

Best regards!

Reviewer 2 Report

Only one:

Line 94 - in my opinion it should be (minimum 1,846 ppb....

Author Response

Dear Reviewer 2,

Thanks a lot for reading our manuscript carefully and your feedback. We have checked the line 94 and are sure that we are referring to the maximum concentration.

Best regards!

Reviewer 3 Report

In this manuscript, the authors showed occurrence of 28 secondary metabolites in 1141 pig feed samples obtained globally. Additionally, the authors showed the cytotoxic activities. These data are clear. I leave some comments on the manuscript. I believe that the following comments will be helpful to improve the manuscript.

Comments

- Did the cytotoxicity mean cell death induction and/or cell cycle arrest induction? If you have any observations (such as microscopic morphology, DNA fragmentation, or others) about the toxicity, please add the results and/or discussion.

- Could you add the data (as a row in Table 2) how many pig feed samples exceeded IC50s determined in the manuscript?

- In Table 2. IC50 values of BEA, I think that “1.905” and “1.756” are typos. Are they correct? “1,905” and “1,756”?

Author Response

Dear Reviewer 3,

thanks for your effort and valuable suggestions for improvement. We really appreciate your comments and refer to them in the text below:

- Did the cytotoxicity mean cell death induction and/or cell cycle arrest induction? If you have any observations (such as microscopic morphology, DNA fragmentation, or others) about the toxicity, please add the results and/or discussion.

We used the sulforhodamine B assay (SRB) for assessing the cytotoxicity of each compound. SRB is a fluorescent protein dye, which binds to the protein content of cultured cells. Results are linear to the cell number and can be extrapolated to measure cell proliferation.

- Could you add the data (as a row in Table 2) how many pig feed samples exceeded IC50s determined in the manuscript?

We have already stated this information in the conclusion (row 375): the maximum analysed concentrations of apicidin, bikaverin and aurofusarin exceed their IC50 values. Apicidin occurs in a concentration of up to 1,568 µg/kg with an IC50 value of 324 µg/kg, bikaverin with an IC50 value of 287 µg/kg was found at a maximum of 1,564 µg/kg and the maximum concentration measured of aurofusarin was 85,360 µg/kg with an IC50 value of 6,766 µg/kg.

- In Table 2. IC50 values of BEA, I think that “1.905” and “1.756” are typos. Are they correct? “1,905” and “1,756”?

Thanks for reading our manuscript carefully! Yes, it was a typo and we corrected it in the table.

Best regards!